# Farmers' Intention to Adopt Agronomic Biofortification: The Case of Iodine Biofortified Vegetables in Uganda

Nathaline Onek Aparo [1,2,*,†], Solomon Olum [3,†], Alice Onek Atimango [1,2], Walter Odongo [2], Bonny Aloka [4], Duncan Ongeng [3], Xavier Gellynck [1] and Hans De Steur [1]

[1] Division of Agri-Food Marketing and Chain Management, Department of Agricultural Economics, Ghent University, Coupure Links 653, 9000 Ghent, Belgium; hans.desteur@ugent.be (H.D.S.)
[2] Department of Rural Development and Agribusiness, Gulu University, Gulu P.O. Box 166, Uganda
[3] Department of Food Science and Postharvest Technology, Gulu University, Gulu P.O. Box 166, Uganda
[4] Department of Science and Vocational Education, Lira University, Lira P.O. Box 1035, Uganda
* Correspondence: nathalineonek.aparo@ugent.be
† These authors contributed equally to this work.

**Abstract:** Agronomic biofortification, the application of fertilizer to increase micronutrient concentrations in staple food crops, has been increasingly promoted as a valuable approach to alleviate micronutrient deficiencies, but its success inevitably depends on farmers' acceptance and adoption. By using iodine fertilizers as a case, this study aimed to understand vegetable farmers' intentions to adopt agronomic biofortification. Therefore, the focus is on the potential role of socio-psychological factors, derived from two well-established theoretical models in explaining adoption intentions. Data from a cross-sectional survey of 465 cowpea and cabbage farmers from a high-risk region of Uganda were analyzed using binary logistic regression. The findings show that 75% of the farmers are likely to adopt agronomic iodine biofortification and are willing to devote a substantial part of their land to this innovation. Farmers' intention to adopt strongly depends on their attitude and control beliefs regarding iodine biofortification, vegetable type, access to extension services, and farmland size. This study highlights the crucial role that behavioral and attitude factors play in communities at risk for nutritional disorders' potential acceptance and sustained implementation of vegetable biofortification. To reinforce the observed positive inclination towards iodine biofortification among vegetable farmers, it is essential to increase awareness of the benefits, potential risks, and consequences of iodine deficiency, accompanied by motivational strategies to enhance farmers' inherent beliefs in their ability to implement this innovation.

**Keywords:** agronomic iodine biofortification; micronutrient malnutrition; behavioral intention; iodine-enriched vegetables; smallholders; Uganda

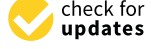



## 1. Introduction

One of the most urgent issues with global development is still hunger and malnutrition, along with the adverse effects they have on health and education [1,2]. Iodine deficiency is one of the most common micronutrient deficiencies, resulting in insufficient thyroid hormone secretion, which causes a variety of iodine deficiency disorders (IDDs). Globally an approximated two billion people—accounting for 30–38% of the world's population—are estimated to have insufficient iodine intake [3]. Due to a lack of recent survey data, Uganda's current prevalence of iodine (I) deficiency is mostly unclear [4], similar to other nations in Africa [5]. Iodine deficiency is considered to be a public health issue at a goiter prevalence of 5% or above [6]. According to a USAID-funded initiative called the food and nutrition technical assistance (FANTA II) research, the total goiter rate (TGR) among school-going children between the ages of six and 12 in Uganda was 31% [7], which is significantly higher than the 5% cut off. Additionally, current stakeholder surveys show that goiter cases are frequently seen in the neighborhood [8]. The high incidence of IDD

has typically been explained by the insufficient amounts of I in edible plant parts as well as the limited availability of I-rich foods like marine foods [9].

Dietary supplements are a good way to combat the drawbacks of iodine deficiency in people. In many nations, the iodine shortage has been reduced thanks to iodine prophylaxis based on the iodization of table salt [10,11]. IDDs, however, continue to be widespread and serious in many rural households in developing nations that are not served by the salt iodization program and have low incomes [12]. Additionally, the implementation of laws intended to restrict salt consumption to avoid hypertension and cardiovascular diseases [13], as well as the instability and susceptibility to vaporization loss of iodine in salt added to food products [14], represent some of the challenges associated with salt iodization. A potential and complementary strategy to combat IDDs and other forms of micronutrient malnutrition is to biofortify plant-based staples such as vegetables [15,16]. Since they contain essential phytochemicals like fiber, vitamins, antioxidants, and minerals, vegetables are a cornerstone of a healthy diet [17]. By enhancing the nutritional value of fruits and vegetables during plant growth as opposed to post-harvest handling and storage, the biofortification of vegetables increases iodine consumption in humans [3,18]. Currently, iodine-rich vegetables are recommended as part of the human diet to alleviate the problem of Iodine Deficiency Disorders (IDDs) and improve the immunity of consumers [19,20]. To achieve this, agronomic biofortification using iodine fertilizers during crop cultivation is being implemented [21]. Iodine fertilizers are known to improve the absorption and accumulation of this essential yet scarce micronutrient, making it readily available in plant-based foods. Foliar application of the fertilizer is highly preferred since it requires much less fertilizer than soil application [22–24]. Previous studies have demonstrated the efficacy of leafy vegetables to absorb and accumulate iodine through fertilizer application [25,26]. Field trials on cabbage and cowpeas in the Gulu and Lira districts revealed that foliar application of I-enriched fertilizers under farmer field conditions increased I concentrations by up to 109.1 mg kg$^{-1}$ at 15 kg I ha$-$ and (5854.2 mg kg$^{-1}$) in cabbage and cowpea respectively [9]. This finding suggests that cowpea and cabbage can be effectively biofortified through foliar application of both KI and KIO$_3$. Furthermore, a projective analysis among Ugandan vegetable farmers demonstrated iodine agronomic biofortification as a highly profitable effort, generating average benefit-cost ratios (BCRs) of 3.13 and 5.69 for cabbage and cowpea production, respectively, higher than the conventional production practice [27]. The same study also highlighted consumers' willingness to pay for iodine-rich cabbage and cowpeas.

In evaluating the importance of agronomic biofortification, it is imperative to consider its adoption by farmers and acceptance by the final consumers of biofortified foods. This viewpoint is because both stakeholders (consumers and producers) significantly influence the adoption of food innovations [28]. A large body of literature has examined the influence of cognitively linked factors, such as attitude and perceptions on the (potential) adoption of biofortified food products among consumers in developing countries with high micronutrient deficiency prevalence rates [29,30]. Despite the fact that the benefits of biofortification in terms of nutrition, cost-effectiveness, and socioeconomic impact are widely documented in the literature, information regarding farmers' perceptions of biofortification remains scarce [31–33].

While supplementation and industrial fortification are nutritional approaches that add minerals and vitamins to food products or as dietary supplements, biofortification is an agriculture-based strategy [34,35]. Consequently, agronomic biofortification implementation must start at the input level where farmers are [36]. The main challenge with biofortification is ensuring that farmers extensively adopt it after variety or fertilizer development [16]. This perspective suggests that consumer acceptance of biofortified foods alone is insufficient to guarantee the success of biofortification. Achieving the anticipated health impacts (reduced micronutrient deficiency) is inextricably linked to farmers' perceptions of and actual adoption of agronomic biofortification [34]. Research highlights that farmers' adoption and acceptance of agronomic iodine biofortification are essential for a participatory, effective, and sustainable implementation of agronomic biofortification

because they are both the primary producers and consumers of biofortified crops [36]. Thus, it is crucial to have a comprehensive understanding of the factors that influence farmers' decisions to use iodine fertilizers for iodine biofortification.

Adopting biofortification will involve a cognitive process, leading to a motivated decision made by farmers. Additionally, the relationship between farmers' perceptions (and/or attitudes) and potential adoption of iodine biofortification can be best predicted based on a priori theory since iodine-biofortified food crops and iodine fertilizers are not yet available on the market (other than for research purposes) in many developing countries.

In this regard, the current study combined two psychosocial theories, the Theory of Planned Behavior (TPB) and the Health Belief model (HBM), to investigate the intention of Ugandan farmers to adopt iodine agronomic biofortification. Specifically, we assessed the role of (a) psycho-social behavioral factors on farmers' intention towards the application of iodine fertilizers, and (b) farm and farmer characteristics on farmers' adoption of agronomic iodine biofortification.

In various respects, this work advances the state of the art. Firstly, compared to the vast body of literature on consumer attitudes, there are very few studies on farmers' perceptions of biofortification. Secondly, it focuses on iodine biofortification, a type of biofortification that is hardly studied in the behavioral research field, compared to vitamin A, zinc, and iron biofortification. Thirdly, the study uses a sizable sample of 465 farmers to focus on a high-risk area for iodine deficiency. Lastly, our model is based on a synthesis of two well-known theoretical models of behavioral change, enabling evaluation of its usefulness in terms of, for example, significant determinants.

## 2. Theoretical Framework

The TPB builds upon the well-known Theory of Reasoned Action (TRA) [37]. TRA hypothesizes that human behavior is a function of intention, which is determined by an individual's attitude towards the specified behavior and subjective norms [38,39]. Attitude refers to an individual's positive or negative assessment of a specific behavior, and is derived from behavioral beliefs [40]. Subjective norms (SN) refer to the perceived social pressure to perform or not perform a behavior based on the expectation of significant others [41]. The TPB was birthed as an extension of the Theory of Reasoned Action by utilizing the two original predictors (attitude and subjective norms), with the addition of a third predictor (perceived behavioral control (PBC)) to explain behavior where barriers to acting exist [38]. PBC is the degree of confidence or control individuals feel to perform a specified action [42]. In the current study, PBC refers to farmers' assessment of their ability to apply iodine fertilizer in crop production.

On the other hand, the HBM posits that two sets of factors influence a person's health behavior: (1) their desire to avoid an illness (e.g., iodine deficiency) or get well in case they are already sick and (2) the belief that a recommended health intervention (e.g., iodine biofortification) will effectively prevent or cure the disease [43–45]. To this end, the HBM incorporates a motivational component and a set of related core beliefs [46]. Regarding motivation, the model suggests that individuals are most likely to perform a recommended health behavior if they have a motivating force or rationale and hold specific beliefs. Furthermore, individuals must believe that (1) they are at risk of developing a health problem or illness (e.g., iodine deficiency disorders) (perceived susceptibility) or that they are ill already; (2) the effect of the disease on their health and social life could be severe (perceived severity); (3) the recommended health action will be feasible, beneficial, and effective at minimizing the seriousness of the health problem; and (4) the recommended preventive action is associated with more benefits than barriers [45,46]. Accordingly, the core constructs of the HBM model are perceived severity, perceived susceptibility, perceived benefits, perceived barriers, and cues to action [47,48]. Furthermore, knowledge about the health condition under investigation and the proposed health behavior to prevent or improve the situation is usually added to the HBM [49,50].

Although they have been applied single-handedly in different contexts [51–54], several authors [39,49] have observed that applying a combination of TPB and HBM (Figure 1) to explain nutrition-related behavior increases their predictive ability, broadens understanding, and enables more variance of outcomes to be explained. Therefore, this study investigated the influence of the constructs of the two theoretical models on farmers' intention to apply iodine fertilizers to vegetable production in Uganda. Furthermore, based on the consideration that the surveyed farmers are largely subsistent, it was necessary to use more consumer-oriented models to study farmers' adoption of agronomic biofortification. Consequently, both models (TPB and HBM) have wide applications in consumer studies on novel food products [55,56]. Figure 1 represents the model integrating the constructs of the TPB and HBM and the proposed relationships between the model constructs.

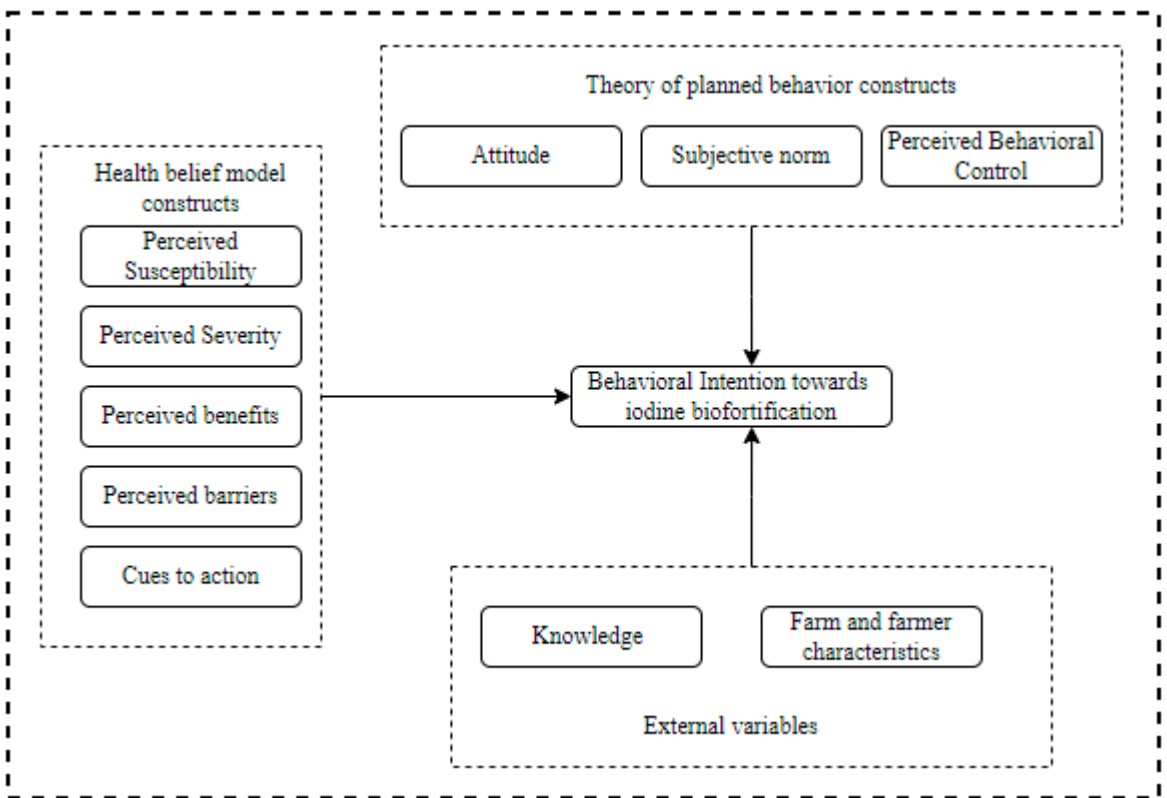

**Figure 1.** An integrated Theory of Planned Behavior and Health Belief Model for the prediction of farmers' behavioral intention towards agronomic iodine biofortification (own compilation based on [37,39,43,45]).

## 3. Materials and Methods

### 3.1. Research Setting

The study was conducted in the Gulu and Lira districts in Northern Uganda (Figure 2). This region has the highest prevalence of malnutrition, especially micronutrient inadequacy, per the Uganda national demographic and health census [8,57]. Moreover, the area is predominantly rural and is home to a majority of poor smallholder farmers who practice agriculture on micro-nutrient deficient soils. A further risk factor for low iodine consumption and IDDs is the area's remoteness from waterbodies, which limits access to iodine-rich foods like fish [7]. Finally, the districts of Gulu and Lira were chosen because agronomic iodine biofortification experiments there revealed that cabbages (*Brassica oleracea*) and cowpeas (*Vigna unguiculata*), which are widely grown and consumed in the region, are effective at absorbing iodine applied as foliar fertilizer [9]. Cowpeas and cabbages are of great importance to the nutrition of rural households in Northern Uganda, where the diets predominately consist of starchy foods, including cassava, maize, sorghum, and millet.

These starchy foods have low nutrient density, making them poor sources of micronutrients such as iodine [58,59]. On the other hand, cowpea is a good source of plant proteins [60], while cabbage is a regular vegetable salad in many Ugandan diets.

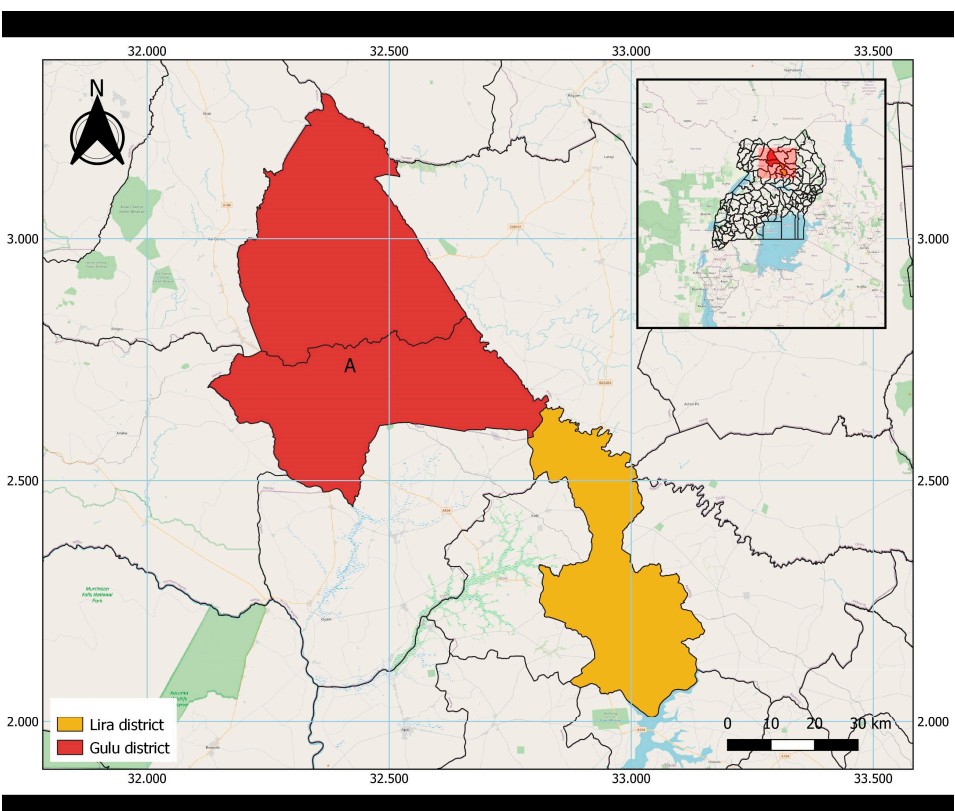

**Figure 2.** A map showing the study areas. A is the present-day Omoro district, which was a part of the Gulu district at the study time.

### 3.2. Sample and Data Collection

The sample consisted of 465 cabbage and cowpea farmers proportionately selected from vegetable production sub-counties in the study districts. Two sub-counties were chosen from each district using simple random sampling from a list of vegetable-producing sub-counties. The farmers to be surveyed were sampled conveniently, as vegetable farmers are fewer than cereal and other crop producers. Only farmers aged 18 years and above, and who provided prior consent, were included in the study.

During data collection, a pre-tested structured questionnaire consisting of three sections was administered face-to-face to each farmer, as the majority had a low level of education. The first section of the questionnaire elicited information on farmers' socio-demographic and farm-related characteristics. The farmers' perceived or subjective knowledge about IDDs and iodine fertilizer was obtained in the second section of the questionnaire. Finally, the last section of the questionnaire contained opinion statements investigating the farmers' intention to apply iodine fertilizer to vegetable production based on the constructs of TPB and the HBM.

### 3.3. Measurement of TPB and HBM Constructs

Farmers were requested to indicate their level of agreement on a five-point scale ("1" strongly disagree and "5" strongly agree) to the following constructs of the TPB and HBM: attitude (ATT), subjective norms (SN), perceived behavioral control (PBC), perceived susceptibility (PSus), perceived severity (PSev), perceived benefits (PBen), perceived barriers (PBar), and cues to action (Cues). Based on earlier research [34,39,49,54,55,61], measurement items for all constructs were formulated and refined using a pre-test in the research

region. Additionally, the scales for negatively phrased questions were reversed to allow for accurate statistical interpretation. For a list of the precise measurement elements, see Supplemental Table S1. Four yes-or-no questions were used to gauge respondents' knowledge about IDDs and iodine fertilizer.

The farmers' intention to use agronomic iodine biofortification was evaluated by first asking the participants if they intended to use iodine fertilizer in their vegetable production (yes/no). They were then asked to state how much land they would commit to producing the vegetables using iodine fertilizer. Measuring intention to adopt by investigating how much resources (land) farmers are willing to put into implementing iodine biofortification is particularly important as most households in developing countries consume what they produce. For example, Schnurr, et al. [36] showed that the potential producers of vitamin A GM biofortified bananas would cultivate the crop both as a staple food and a cash crop, as already done for conventional bananas, which further shows the need to investigate their commitment to invest resources in biofortified food production. Oparinde, et al. [34] applied the same approach in their study on farmers' intention to cultivate Provitamin A genetically modified (GM) cassava in Nigeria. The authors asked farmers to state the percentage of their cassava land area they would be willing to dedicate to the cultivation of Provitamin A GM cassava.

### 3.4. Statistical Analyses

The main evaluation instrument for this study was the Statistical Package for Social Sciences (IBM SPSS statistics) version 23. The items from the primary questionnaires were pre-screened and cross-checked for missing data and outliers as a precondition for the subsequent analysis. To investigate the socioeconomic and agricultural characteristics of the farming households, descriptive statistics were utilized. Multiple item constructs of the integrated TPB and HBM model were examined for reliability and internal consistency using Cronbach's alpha and the item-total correlation [62]. If excluding an item from a construct significantly increased the Cronbach's $\alpha$ of the construct, and in case there was a low item-total correlation (<0.3), the item was removed from that construct [61]. As shown in Table 1, the Cronbach's alpha for each factor was more than the suggested value of 0.6 [63]. The lowest tolerance value was 0.591 and the Variance Inflation Factors (VIF) ranged between 1.006 and 1.882, which is consistent with the maxim that tolerance values greater than 0.1 and VIF values less than 5 suggest the absence of multicollinearity among constructs [64]. The link between the model constructs and the behavioral intention to implement agronomic iodine biofortification was examined using binary logistic regressions. There were no appreciable differences between the farm sites and the target crops after evaluating the models independently for cabbage and cowpea farmers in the two districts. Thus, results are presented and discussed for the overall sample rather than sub-groups.

**Table 1.** Statistics of Construct Reliability.

| Construct | Number of Items | Tolerance | VIF | Cronbach's Alpha ($\alpha$) | Constructs | Number of Items | Tolerance | VIF | Cronbach's Alpha ($\alpha$) |
|---|---|---|---|---|---|---|---|---|---|
| ATT | 5 | 0.591 | 1.692 | 0.718 | PSev | 4 | 0.59 | 1.695 | 0.771 |
| SN | 4 | 0.994 | 1.006 | 0.628 | PBen | 3 | 0.726 | 1.377 | 0.748 |
| PBC | 5 | 0.594 | 1.684 | 0.831 | PBar | 3 | 0.815 | 1.228 | 0.732 |
| PSus | 5 | 0.641 | 1.56 | 0.804 | Cues to Action | 3 | 0.972 | 1.029 | 0.752 |

VIF = Value Inflator Factor; PBen = Perceived benefits; ATT = Attitude; PBC = Perceived behavioral control; PSev = Perceived severity; SN = Subjective norms; PSus = Perceived susceptibility and PBar = Perceived barriers.

## 4. Results

### 4.1. Sociodemographic Profile of Respondents

The characteristics of the 465 farmers who were interviewed are presented below. As indicated in Figure 3, most of the participants owned their farmland and had at least a

primary level of education. Additionally, 59% and 76% of the farmers interviewed were male and had the intention of adopting iodine agronomic biofortification, respectively. The average age of the participants was 38.1 years. At the same time, the farming households consisted of an average of seven members with a modest household monthly income of 331,558 Ugandan shillings (about 92 USD) (Table 2). Still, the income levels were highly variable among the participants.

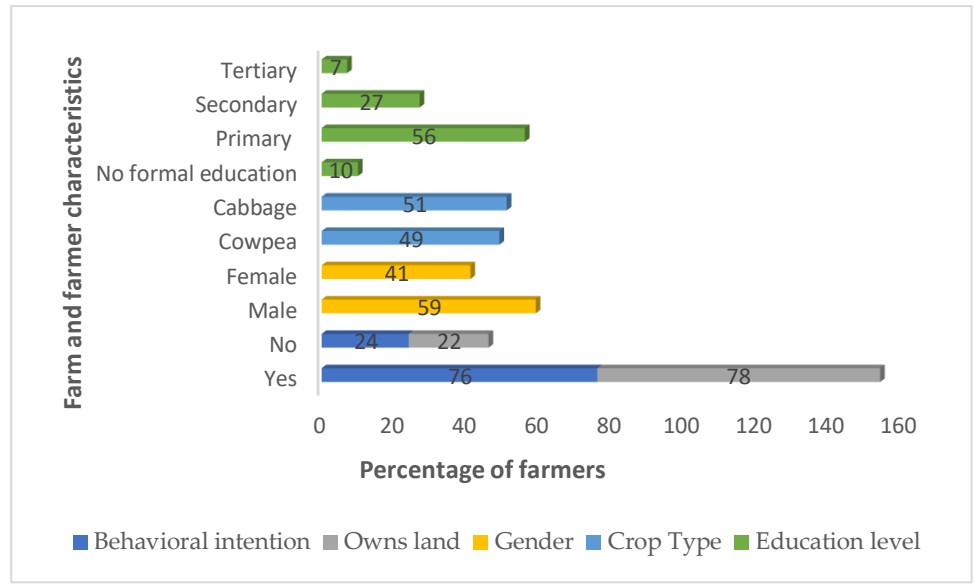

**Figure 3.** Percentage of farmers and their sociodemographic characteristics.

**Table 2.** Farmers' Socio-Demographic Characteristics.

| Characteristic | Mean | SD |
|---|---|---|
| Age (complete years) | 38.1 | 12.6 |
| Household size | 7 | 3.0 |
| Household income (UGX.) | 331,558 | 388,829 |
| Farm size (hectares) | 1.82 | 1.66 |
| Experience applying fertilizers | 3.2 | 3.4 |

Remarks: Only the category that responded "yes" was reported for knowledge, intention to apply iodine fertilizers, and access to extension; IDDs = iodine deficiency disorders, UGX = Ugandan shillings (currency). Concerning the household income, 1USD = 3700 UGX as of the data collection period.

### 4.2. Knowledge about Iodine Deficiency Disorders, Fertilizers and Sources

The majority of the participants did not know any food source of iodine (79.4%), the different forms of iodine deficiency (70.0%), or iodine fertilizers (85.1%) as shown in Figure 4. However, most farmers (76%) indicated an intention to apply iodine fertilizer to bio-fortify their crops during production.

Farmers' perception towards the use of iodine fertilizers to enhance the nutrient content of vegetables was assessed through predetermined opinion statements corresponding to the constructs of the integrated TPB and HBM. The results in Table 3 indicate that the mean scores for statements of most constructs were generally high. These high scores show that many participants agreed and strongly agreed with most statements (3.02 to 4.43). However, the knowledge score was low (1.41–1.95), showing that most participants did not know about iodine fertilizers, food sources of iodine, and iodine deficiency disorders. The poor knowledge regarding iodine fertilizers is attributable to the novel nature of iodine fertilizers and their current absence in the Ugandan market. Farmers who answered yes to this question participated in prior agronomic experiments conducted in the study areas.

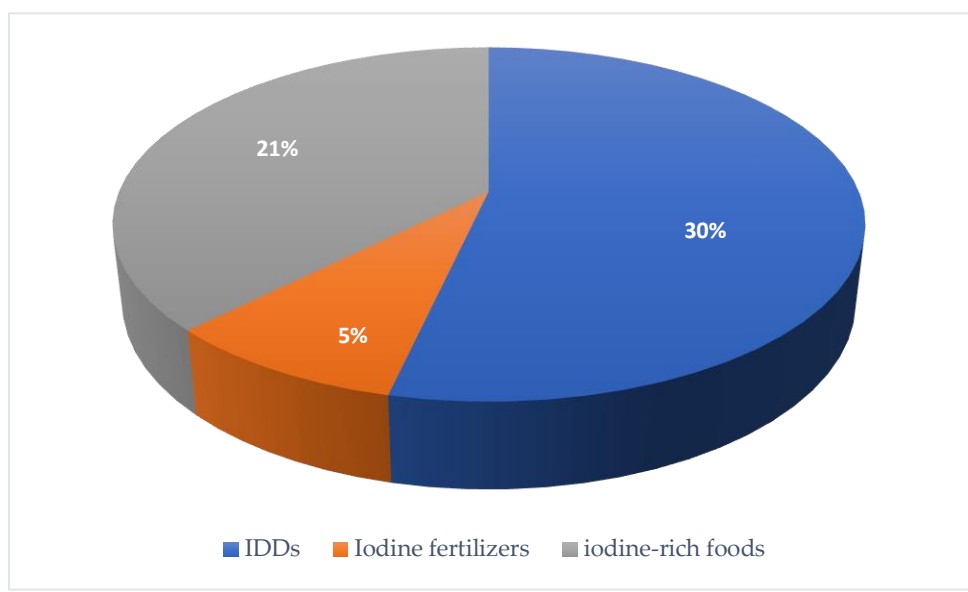

**Figure 4.** Percentage of farmers with knowledge on iodine deficiency, fertilizers, and sources. Farmers' knowledge, perceptions, and intended adoption of iodine agronomic biofortification.

**Table 3.** Descriptive statistics of constructs and items.

| Construct | Mean | S. D | Constructs | Mean | S. D |
|---|---|---|---|---|---|
| Attitude | 3.92 | 0.762 | Perceived severity | 4.41 | 0.621 |
| ATT 1 | 4.35 | 0.844 | PSev1 | 4.19 | 1.022 |
| ATT2 | 4.09 | 0.942 | PSev2 | 4.40 | 0.788 |
| ATT3 | 4.31 | 0.895 | PSev3 | 4.51 | 0.698 |
| ATT 4 | 2.76 | 1.212 | PSev4 | 4.55 | 0.671 |
| ATT 5 | 4.11 | 0.950 | Perceived benefits | 4.43 | 0.580 |
| Subjective norms | 3.02 | 1.112 | PBen1 | 4.48 | 0.710 |
| SN1 | 2.95 | 1.757 | PBen2 | 4.47 | 0.688 |
| SN2 | 3.12 | 1.660 | PBen3 | 4.36 | 0.736 |
| SN3 | 2.97 | 1.643 | Perceived barriers | 4.16 | 0.823 |
| SN4 | 3.05 | 1.453 | PBar1 | 4.32 | 0.967 |
| Perceived behavioral Control | 3.87 | 0.946 | PBar2 | 3.98 | 1.115 |
| PBC1 | 3.72 | 1.267 | PBar3 | 4.19 | 0.972 |
| PBC2 | 4.08 | 1.136 | Cues to Action | 4.36 | 0.618 |
| PBC3 | 4.29 | 1.059 | Cues1 | 3.97 | 1.316 |
| PBC4 | 3.46 | 1.389 | Cues2 | 4.50 | 0.799 |
| PBC5 | 3.83 | 1.253 | Cues 3 | 4.63 | 0.630 |
| Perceived susceptibility | 4.09 | 0.714 | Knowledge | 1.68 | 1.037 |
| PSUS1 | 4.03 | 1.042 | Know1 | 1.70 | 0.459 |
| PSUS2 | 4.17 | 0.903 | Know2 | 1.95 | 0.217 |
| PSUS3 | 4.24 | 0.912 | Know3 | 1.41 | 0.492 |
| PSUS4 | 4.02 | 1.016 | | | |
| PSUS5 | 4.28 | 0.881 | | | |

Remark: all statements were measured on a 5-point Likert Scale (1 = strongly disagree, 5 = strongly agree). PBen = Perceived benefits; ATT = Attitude; PBC = Perceived behavioral control; PSev = Perceived severity; SN = Subjective norms; PSus = Perceived susceptibility and PBar = Perceived barriers.

### 4.3. Determinants of Farmers' Intention to Adopt Agronomic Iodine Biofortification

Four binary logistic regressions, including 465 cases, were performed to determine the factors influencing farmers' intention to adopt agronomic iodine biofortification.

The baseline model, which contains the predictions of the category that occurred most often in the study dataset, was a statistically significant predictor of farmers' adoption intention. Since the proportion of farmers who had intentions to apply iodine fertilizer

(76.3%) was higher than those who had no such intention (23.7%), the baseline model guessed yes consistently and correctly classified 76.3% of the farmers with the intent to apply iodine fertilizers. In addition, the overall model evaluation based on the omnibus test of model coefficient ($p = 0.000 < 0.005$) and Hosmer and Lemeshow goodness of fit statistics ($p > 0.05$) indicated that all four models are statistically significant for predicting and differentiating farmers' intention to apply iodine fertilizers and fit the data used well.

Regarding Model 1 (Figure 5), a three-predictor logistic model was fitted to the data to test the research hypothesis regarding the relationship between a farmer's intention to apply iodine fertilizers and the TPB constructs of attitude, subjective norms, and PBC. This model explained 65% (Nagelkerke $R^2$) of the variance in farmers' intention to adopt iodine biofortification and correctly classified 89.7% of farmers intending to apply iodine fertilizers. Attitude ($\beta = 0.80$; $p < 0.001$) and PBC ($\beta = 0.90$; $p \leq 0.001$) were found to make significant positive contributions to the model and, therefore, to influence farmers' intention to apply iodine fertilizers. On the other hand, the influence of subjective norms on the intention to adopt iodine fertilizers was insignificant.

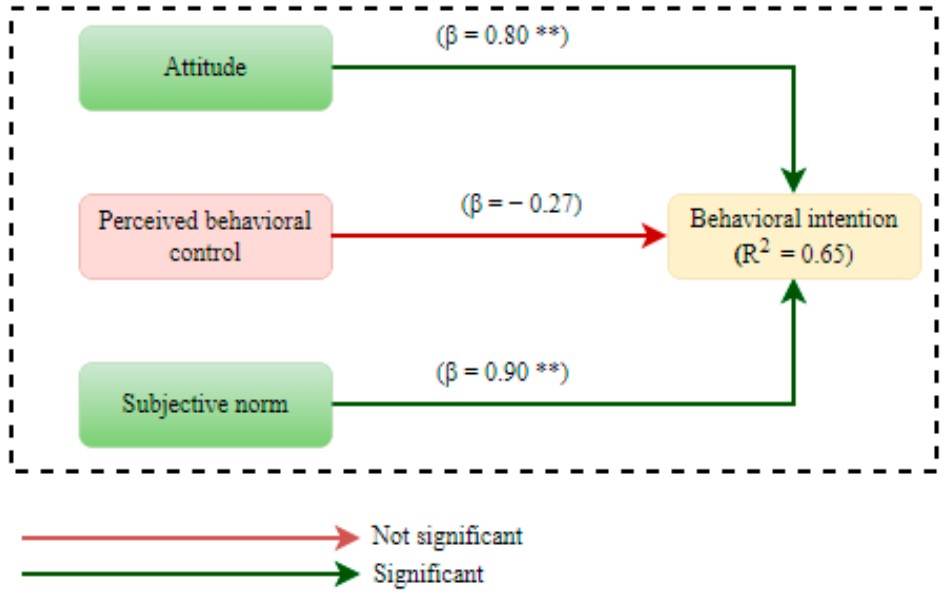

**Figure 5.** Predicting Behavioral intention based on the Theory of Planned Behavior constructs (Model 1). * $p < 0.05$, ** $p < 0.001$, $R^2$ = Nagelkerke R square.

Model 2 (Figure 6) used the HBM constructs (perceived severity, perceived susceptibility, perceived barriers, perceived benefits, and cues to action) to predict the farmers' intention to apply iodine fertilizers to their vegetable crop production. This model correctly classified 76.3% of farmers with the intent and explained only 9% (Nagelkerke R2) of the variance in farmers' intention to adopt iodine biofortification. Out of the HBM variables, only perceived barrier ($\beta = -457$) and perceived benefits ($\beta = 592$; $p = 0.005$) had, respectively, a significant negative and positive effect on the intention to adopt iodine biofortification.

Model 3 (Figure 7), based on the integrated TPB and HBM constructs, had an overall categorization accuracy of 90.1% and explained 70% (Nagelkerke $R^2$) of the variance in farmers' behavioral intention to adopt iodine biofortification. Attitude ($\beta = 0.75$; $p \leq 0.001$) and PBC ($\beta = 0.90$; $<0.001$) are the only factors that had a significant positive influence on predicting farmers' intention to apply iodine fertilizer. Compared to model 2, perceived barriers and perceived benefits became insignificant in influencing farmers' adoption intention regarding agronomic iodine biofortification in model 3.

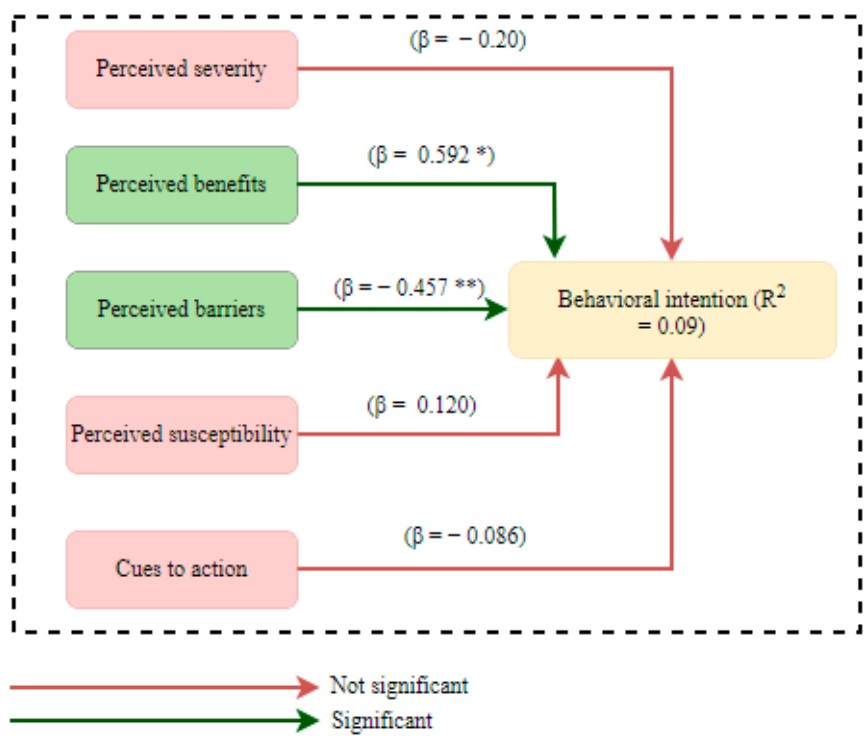

**Figure 6.** Predicting Behavioral intention based on the Health Belief Model constructs (Model 2). $R^2$ = Nagelkerke R square, * $p < 0.05$, ** $p < 0.001$.

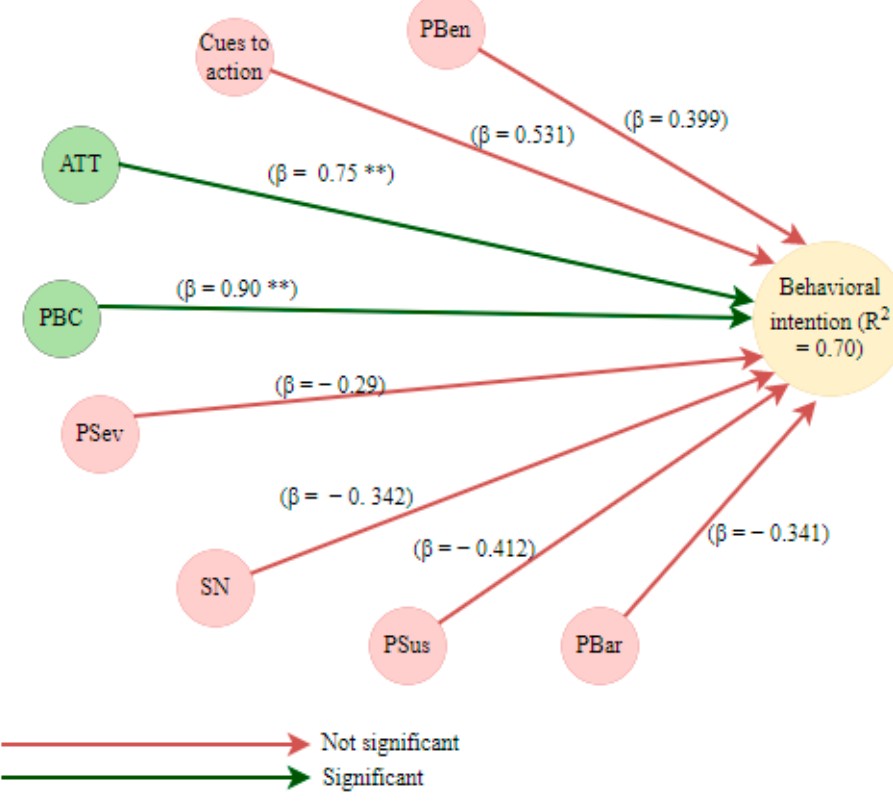

**Figure 7.** Predicting Behavioral intention based on the integrated TPB and HBM theory (Model 3). $R^2$ = Nagelkerke R square, * $p < 0.05$, ** $p < 0.001$. PBen = Perceived benefits; ATT = Attitude; PBC = Perceived behavioral control; PSev = Perceived severity; SN = Subjective norms; PSus = Perceived susceptibility and PBar = Perceived barriers.

Model 4 (Figure 8) looked at the role of external variables, including knowledge and four socio-demographic variables. With a classification accuracy of 76.1%, the model accounted for 14% (Nagelkerke $R^2$) of the variance in farmers' intention to adopt agronomic iodine biofortification. Out of the factors tested, only crop type 1 ($\beta = 0.630$; $p = 0.026$), farmland size ($\beta = 0.101$; $p = 0.028$), and access to extension ($\beta = 0.801$; $p \leq 0.001$) were found to be significant positive predictors of the behavioral intention to use iodine fertilizers by farmers.

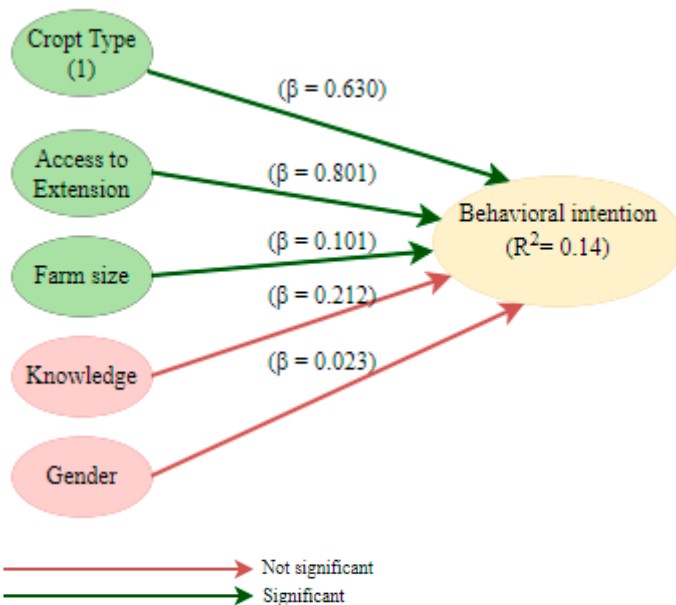

**Figure 8.** Predicting Behavioral intention using the sociodemographic characteristics of farmers (Model 4). Crop Type 1 = Cowpea; $R^2$ = Nagelkerke R square; * $p < 0.05$, ** $p < 0.001$.

## 5. Discussion

The success of nutrition-based interventions like biofortification inevitably depends on farmers' acceptance and adoption of it. Understanding the intentions of farmers and the factors that influence them thereof allows for the needs of other stakeholders including food industries, retailers, and consumers to be met, and facilitates the achievement of the desired health/nutritional impacts of biofortification. Therefore, this study investigated how the constructs of HBM and TPB could predict the intention of Ugandan farmers to adopt iodine agronomic biofortification of food crops, achieved through the application of iodine-rich fertilizers and their opinions on including iodine biofortified vegetables in their family diets.

Attitude had a significant positive relationship with behavioral intention to apply iodine fertilizers. Attitude was reported as a consistent determinant of willingness to adopt or accept biofortification in past studies [34,39,65]. Thus, farmers who positively evaluate iodine biofortification are more likely to accept and adopt the technology.

The significant positive effect of the perceived behavioral control means that farmers who believe that the decision to adopt agronomic iodine biofortification is within their volitional control to cultivate food crops by applying iodine fertilizers could be willing to embrace the iodine biofortification. This effect also highlights that farmers should feel they have control over production resources to cultivate iodine-biofortified foods. This finding corresponds with previous studies that applied the TPB models in novel food consumption or production. For instance, Talsma, et al. [55] found that control beliefs strongly predicted caregivers' intention to give vitamin A biofortified cassava to school children in Kenya, similar to the study of Sulaiman, et al. [66]. It is not surprising that PBC had a favorable impact on intention given that survey respondents had, on average, used fertilizers for crop production for over three years. Farmers may be encouraged to use iodine fertilizers by their prior experience using other chemical fertilizers. The positive influence of PBC

also implies that extension services tailored towards improving the capability of farmers to apply iodine fertilizer and manage the production process, including acquiring inputs such as iodine fertilizer and seeds, might increase the likelihood of adoption.

It was anticipated that social pressure to biofortify vegetables by using iodine fertilizer would positively affect behavioral intention [66]. However, subjective norms did not significantly predict farmers' intentions in this study. This finding is consistent with that of Talsma, et al. [55] who reported a non-significant relationship between subjective norms and intention to consume pro-vitamin A-rich cassava among primary school children in Kenya. This result may be explained by the fact that iodine biofortification was not yet being implemented and communicated in Uganda when the study was conducted. A stronger influence of social pressure would be expected if the significant referents of the farmers were already using the technology or if they knew much about it.

Within the group of external variables, access to extension, crop type, and farm size were found to affect the intention of farmers to apply iodine fertilizers. While the literature on the effect of farm size on adoption has been inconclusive, the significant positive impact of farm size on farmers' adoption of new technology has been reported in previous studies. In the context of this study, land is a valuable asset in the Northern part of Uganda as it is in Uganda generally. Farm size is associated with the kind of production one engages in, i.e., subsistence or commercial. Therefore, people with smaller farms might find it hard to dedicate a portion of it to agronomic iodine biofortification. They would instead use it to produce the more traditional staples, including millet, cassava, and beans. However, farmers with extensive farmland are shown to be more willing to allocate some of it for agronomic iodine biofortification.

Regarding crop type, a change in crop choice from cabbage to cowpea would result in 0.507 higher odds for the intention to apply iodine fertilizer. This difference might be due to the low cost associated with cowpea production relative to cabbage, which translates into a higher benefit-cost ratio for iodine cowpea [27]. Even though cabbages are eaten all over Uganda, cowpea, locally referred to as "boo," is culturally produced and consumed in Northern Uganda in larger proportions compared to other regions by smallholder farmers. A total of 90% of cowpea production is from Uganda's Northern and Eastern regions. Based on this finding, future vegetable iodine biofortification initiatives may focus on cowpea farmers initially since they are more likely to embrace the innovation.

The results show that the two predictors of intention in the integrated model (PBC and attitude) belong to TPB. The HBM constructs were less valuable as predictors of intention to adopt iodine biofortification. The model based on TPB constructs accounted for more (65%) variance in intention than model 2, which accounted for only 9% of the variance in farmers' intention to adopt agronomic iodine biofortification. This finding conforms with previous studies that have found that TPB constructs explained more variance in behavioral intention than HBM when the two theories are applied together [61]. In their study on the cultural acceptability of biofortified sweet potatoes using the combined TPB and HBM, Hummel, et al. [61] found that only subjective norms and attitudes (both belonging to the TPB) significantly predicted consumption among households with children in Malawi. There are possibly two explanations for the dominance of the TPB model over HBM in predicting intention to adopt iodine agronomic biofortification. Firstly, intention is initially not part of the HBM [47], even though it has been previously used in combination with other models, mainly when predicting behavioral intention towards novel foods and novel food technologies through the use of TPB [39,49,55]. As such, the ability of HBM to predict behavioral intention on biofortified food production needs to be separately validated. Secondly, the participants in the current study, who were farmers, may have considered more the production attributes of iodine biofortification when indicating their intention to adopt and less the consumption attributes. Therefore, the economic drivers could have outweighed the health drivers (perceived severity and perceived susceptibility) in predicting farmers' adoption intention, although they also cultivate for their own consumption.

Overall, the combined model predicted a higher variance in intention to adopt iodine agronomic biofortification than when TPB and HBM were applied separately. The low predictive power of the constructs of HBM in the combined model shows that a more significant variation in the intention to adopt iodine biofortification can be predicted by other factors which were not considered in this study. However, given that decision-making is a process, the stated intention of farmers to adopt iodine biofortification can be regarded as an immediate reaction to learning about the innovation. While many respondents intended to adopt this iodine biofortification, their behavioral intentions may change over time as farmers know more about the innovation. Therefore, future studies could investigate how the behavioral reactions will change when more information becomes available and awareness increases.

## 6. Study Limitations

Despite having a solid theoretical foundation and providing insights on factors crucial for farmers to embrace and adopt agronomic iodine biofortification, the generalizability of our study findings is subject to certain limitations. Firstly, neither iodine fertilizers nor iodine biofortified cabbages/cowpeas were available on the Ugandan market at the time of the study. As such, the farmers surveyed had neither used iodine fertilizers nor consumed iodine-rich vegetables. Therefore, it was impossible to examine the relationship between behavioral intention and actual behavior. It might be interesting for future research to assess how behavioral intention would change over time and influence farmers' actual behavior regarding agronomic iodine biofortification.

Secondly, our study was conducted in a limited geographic context of two districts in Northern Uganda. Assessing the validity of this model while incorporating other factors with farmers from various cultural backgrounds in industrialized and emerging countries would be theoretically and practically valuable because the factors that influence technology adoption may vary from place to place.

Thirdly, we employed a cross-sectional design by which data were collected at only one point. However, further research would allow a better understanding of whether the significance of the constructs might alter with time or in different circumstances. For example, the influence of subjective norms, barriers, perceived severity, perceived susceptibility, and cues to action might become significant when (1) the iodine fertilizers and iodine-rich vegetables become available on the Ugandan market, (2) information about the technology and its benefits are being shared, and (3) government and other stakeholders take affirmative actions towards the implementation of such an innovation.

Fourthly, Fanou-Fogny, et al. [67] noted that self-reports on behavioral intention, as in our study, could be influenced by social desirability/approval, especially where the desired response is obvious to the participant. Talsma, et al. [55] asserts that respondents may occasionally find it difficult to disagree with statements made to them and end up agreeing with nearly all of them.

## 7. Conclusions

The study aimed at determining the predictors of intention to adopt agronomic iodine biofortification among cowpea and cabbage farmers in Uganda to gain acumens into how the two vegetables can be successfully biofortified by applying iodine fertilizer in the future. While the HBM could predict only a slight variance in behavioral intention, the relevance of some of the critical factors of both TPB and HBM was established.

In general, most farmers were willing to dedicate some portion of their vegetable farmland to implement iodine biofortification, which is a positive result for the future deployment of this intervention. The significance of perceived behavioral control, attitude, and perceived severity shows that the promotion of iodine biofortification could target motivating farmers to have positive attitudes towards the intervention and the biofortified food products through sensitization about agronomic biofortification and its benefits. Secondly, farmers should feel they control production processes, including allocating

resources such as land. In order to increase the consumption of iodine-biofortified foods once they are made available, farmers should believe that they have control over what is eaten in their homes and can therefore include the iodine-rich vegetables they grow in their family's diet. Finally, a focus on educating farmers about the advantages of iodine biofortification should be made. This will alter their perspective of the technology and could lead to wider adoption.

Although more research is needed to understand the motivation of farmers to adopt iodine agronomic biofortification, this study has demonstrated the role of the cognitive-based constructs of the HBM and TPB in predicting their adoption intention. It has further shown high potential for adoption of the technology based on behavioral intention to adopt.

**Supplementary Materials:** The following supporting information can be downloaded at: https://www.mdpi.com/article/10.3390/horticulturae9030401/s1, Table S1: Constructs used in the study, Table S2: Predicting farmers' intention to adopt iodine agronomic biofortification.

**Author Contributions:** N.O.A., S.O. and H.D.S. conceptualized and designed the study, analyzed, drafted, reviewed and edited the article. In addition, N.O.A. and S.O. conducted the study, A.O.A., B.A., W.O., D.O. and X.G. reviewed and edited the manuscript as well as helping with the discussion and interpretation. In addition, H.D.S. and D.O. supervised the study. All authors have read and agreed to the published version of the manuscript.

**Funding:** This study was conducted within the framework of the VLIR-TEAM project "Agronomic iodine biofortification to increase iodine intake in Northern Uganda: a stakeholder-based intervention", funded by the Flemish inter University Council (VLIR) (ZEIN2016PR429).

**Informed Consent Statement:** Informed consent was obtained from all subjects involved in the study.

**Data Availability Statement:** Data can be provided when requested.

**Conflicts of Interest:** The authors declare no conflict of interest.

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
