# Peer review of "Farmers’ Intention to Adopt Agronomic Biofortification: The Case of Iodine Biofortified Vegetables in Uganda"

_horticulturae, doi:10.3390/horticulturae9030401_

Round 1
Reviewer 1 Report
Dear Authors,
the manuscript is interesting and well-written, however, it needs some corrections:
1. please check punctuation and spaces;
2. in "Materials and Methods" subsection 3.2. - please rewrite the part with pre-questions, it is confusing. First present questions, then explain the conditions of the questionnaire;
3. results presentation - a graphical/visual presentation would be more suitable than dry numbers in tables, please add diagrams.

Reviewer 2 Report
Keywords: Please reduce the number of keywords to 6
Line 72-73 “Yet, unlike other micronutrient deficiency interventions such as supplementation and industrial fortification, biofortification is an agriculture-based intervention (Kamran & Asif, 2012; Kent, 2014).” it is not clear. Please explain
Line 91-100 Aim of this study is not clean. There is no defined and clearly defined research goal. The research presented in the reviewed manuscript is a kind of report on the situation of plant biofortification in iodine. The authors try to answer the question whether farmers will be interested in biofortification or not. Therefore, the manuscript on the nature of economic considerations should be published in another journal.
Table 2 please change Farm size (acres) to Farm size (hektar)
Authors should revise the Literature section as required for authors
The reviewed manuscript is very interesting and concerns the problems of plant biofortification in iodine. It is one of the four elements that are deficient worldwide. In my opinion, the work is a report and research of an economic nature, not applied research involving the study of the effect of new fertilizers on improving the biofortification of plants in iodine. Therefore, I believe that the work should not be published in Horticulturae but in another journal with an economic profile. I'm sorry, but I have to reject the article.
Reviewer 3 Report
Major comments
1) Indicate benefits of cowpea and cabbage biofortification with iodine. Levels of biofortification, increase in product quality and yield. (you may use the data from [Ojoke et al, 2019]). Effect on yield is especially important for farmers when economics prevails above health drivers.
2) Taking into account the international character of publication and the importance of IDD protection in many countries, it is highly desirable to indicate more clearly the problem of iodine deficiency in Uganda providing appropriate discussion and concrete citations, such as:
(i)Bimenya GS, Olico-Okui, Kaviri D, Mbona N, Byarugaba W. Monitoring the severity of iodine deficiency disorders in Uganda. Afr Health Sci. 2002 Aug;2(2):63-8.
(ii) Atukunda et al, 2021, The association of urine markers of iodine intake with development and growth among children in rural Uganda: a secondry abalysis of a randomized education trial//Public Health Nutrition, 24(12), 3730-3739. Doi:10.1016/S1368980020001603
3) add geographical coordinates and a map of Gulu, Lira districts in Uganda
Minor comments:
1) Use author guidelines for the whole text: intends, citations, references in the reference list.
2) Thus, don’t use Italics for Tables and their titles. ‘Table’ should be written by bold letters
3) Don’t use Italics for Table notes
4) Citation in the text should be given according to: [1], [2], etc
5) Reference list should be composed according to the authors guidelines. For instance:
Ref. ‘Abizari, A. R., Pilime, N., Armar-Klemesu, M., & Brouwer, I. D. (2013). Cowpeas in Northern Ghana and the Factors 411 that Predict Caregivers' Intention to Give Them to Schoolchildren. PLoS ONE, 8(8), 1-8. 412 doi:10.1371/journal.pone.0072087’ should be changed to:
‘Abizari, A. R.; Pilime, N.; Armar-Klemesu, M.; Brouwer, I. D. Cowpeas in Northern Ghana and the factors that predict Caregivers' intention to give them to schoolchildren. PLoS ONE, 2013, 8(8), 1-8. 412. doi:10.1371/journal.pone.0072087’
6) The reference list should not be given in alphabetic order
7) Line, 577, 582: don’t use dots instead of authors names. All authors should be included
8) ’Table 1- decipher VIF in the note under the Table
9) Lines 255, 273: ‘,001’ change comma to dot ‘.001’
10) Table 4- the same
